# Crystal structure of human Mediator subunit MED23

Didier Monté[1], Bernard Clantin[1], Frédérique Dewitte[1], Zoé Lens[1], Prakash Rucktooa[1,4], Els Pardon[2,3], Jan Steyaert [2,3], Alexis Verger [1] & Vincent Villeret [1]

The Mediator complex transduces regulatory information from enhancers to promoters and performs essential roles in the initiation of transcription in eukaryotes. Human Mediator comprises 26 subunits forming three modules termed Head, Middle and Tail. Here we present the 2.8 Å crystal structure of MED23, the largest subunit from the human Tail module. The structure identifies 25 HEAT repeats-like motifs organized into 5 α-solenoids. MED23 adopts an arch-shaped conformation, with an N-terminal domain (Nter) protruding from a large core region. In the core four solenoids, motifs wrap on themselves, creating triangular-shaped structural motifs on both faces of the arch, with extended grooves propagating through the interfaces between the solenoid motifs. MED23 is known to interact with several specific transcription activators and is involved in splicing, elongation, and post-transcriptional events. The structure rationalizes previous biochemical observations and paves the way for improved understanding of the cross-talk between Mediator and transcriptional activators.

[1] CNRS, UMR 8576—UGSF— Unité de Glycobiologie Structurale et Fonctionnelle, Univ. Lille, 59000 Lille, France. [2] VIB-VUB Center for Structural Biology, Pleinlaan 2, 1050 Brussels, Belgium. [3] Structural Biology Brussels, Vrije Universiteit Brussel, Pleinlaan 2, 1050 Brussels, Belgium. [4]Present address: Heptares Therapeutics Ltd., Broadwater Road, Hertfordshire AL7 3AX, UK. These authors contributed equally: Didier Monté, Bernard Clantin. Correspondence and requests for materials should be addressed to D.M. (email: didier.monte@univ-lille.fr) or to V.V. (email: vincent.villeret@univ-lille.fr)

MED23 was originally discovered as a genetic suppressor of a hyperactive ras phenotype in *Caenorhabditis elegans*[1] and identified as the major target of the conserved transactivation region 3 (CR3) of the adenoviral 13SE1A oncoprotein[2]. MED23 belongs to the Tail module of Mediator, whose main function is to connect Mediator to sequence-specific transcription factors, and is thus of fundamental importance[3]. In vitro and in vivo studies have demonstrated a specific interaction between MED23 and the ternary complex factor Ets-like protein-1 (ELK1), which is activated by MAPK signaling[4]. Functional investigation of MED23 and ELK1 in Ras-active lung cancer demonstrated a critical role for MED23 in enabling the Ras-addiction of lung carcinogenesis, providing MED23 as a potential therapeutic target[5]. ELK1 interacts with MED23 in a complex manner involving many phosphorylation sites[6]. MED23 also binds to other transcription factors, such as ESX (ESE-1/ELF3/ERT/Jen), RUNX2 or IRF7[7–9]. In addition, MED23 plays important roles in postrecruitment steps. It regulates alternative mRNA processing via direct interactions with splicing factors[10], promotes Pol II into transcription elongation via a direct interaction with CDK9 in P-TEFb[11], or promotes histone post-translational modifications on active genes[12]. Altogether, these data underlie the important roles of MED23 in the human Mediator complex and its link with several pathologies. Results obtained from structural biology efforts provided important information on the architecture of yeast Mediator and defined the location of core Mediator on Pol II. These efforts culminated in the cryoEM and X-ray crystallographic studies of the Head and Middle modules reported at atomic or near-atomic resolutions[13–21]. However, structural information remains limited to core Mediator and data about the Tail module are needed to expand our understanding of Mediator function. Here we report the crystal structure of MED23, the largest subunit from the human Tail module. MED23 is composed of 25 HEAT repeats-like motifs organized into 5 α-solenoids and forming an arch-shaped conformation, with an N-terminal domain (Nter) protruding from a large core region. MED23 interacts with several specific transcription activators and is involved in splicing, elongation, and post-transcriptional events. The structure thus paves the way for improved understanding of the cross-talk between Mediator and transcriptional activators and represents an important step towards the complete structural characterization of the human Mediator Tail module.

## Results and Discussion

**Nanobody-assisted crystallization of MED23.** Despite many attempts, recombinant human MED23 could not be crystallized alone. We thus generated specific lama nanobodies, one of which, Nb106, assisted in the crystallization process and allowed us to obtain well-diffracting crystals (see Methods and Supplementary Information). The nanobody binds via its complementarity determining region (CDR) 3 to a cavity in MED23 formed at the junction of three structural regions (named hereafter 5-HEAT, 6-HEAT, and C-HEAT). It is also involved in the crystal packing, with its β-sheet stacking against helices H4, H6, and H8 from the N-HEAT solenoid domain of MED23. We solved the structure of the MED23-Nb106 complex de novo by multiple isomorphous replacement. Diffraction data to 2.8 Å resolution resulted in excellent electron density (Supplementary Fig. 1) and a refined structure that has a free $R$ factor ($R_{free}$) of 24.0% (Supplementary Table 1).

**Overall MED23 architecture.** The MED23 structure shows that the protein is fully folded (with the exception of the last 30 amino acids) and adopts a global form measuring approximately 145 × 55 × 50 Å (Fig. 1a–e). The structure presents an N-terminal domain (N-HEAT: residues 1–220) extending from a core region that encompasses the rest of the protein (Core MED23: residues 221–1334). Core MED23 adopts a compact and flattened triangular form with sides of 80–100 Å in which the protein wraps on itself and positions the C-terminal in close proximity to N-HEAT. Together with N-HEAT, Core MED23 defines an arch of approximately 140 Å (Fig. 1c), with a concave and a convex face (Fig. 1a, d). The structure identifies 71 α-helices encompassing ~65% of the complete sequence (Supplementary Fig. 2). These α-helices engage in 25 heat repeats-like (HR) motifs arranged in 5 right-handed superhelical solenoids. A linker and a bridge domain complete the structure (Fig. 1a–e).

**The N-terminal domain comprises five HR-like motifs.** N-HEAT folds into a solenoid of five HR motifs, with the last four motifs arranged nearly linearly (Fig. 2a). N-HEAT contacts the Core MED23 via helices H10 and H13 which stack against the C-terminal domain. Solenoids composed of HR motifs are known to play a role in mediating protein−protein interactions and in MED23 the two faces of N-HEAT are properly exposed for such interactions. MED23 binds specifically to the RNA Recognition Motif 2 (RRM2) domain of hnRNP-L, a sequence-specific RNA binding protein which is involved in splicing[10]. The region of MED23 involved in this interaction maps to a fragment of MED23 covering residues 1–327 [10], which corresponds to N-HEAT extended up to the linker and the first HR motif of the next solenoid (3-HEAT, defined below). MED23 also physically associates via this region with the heterodimeric E3-ubiquitin ligase RNF20/40 which functions with the super elongation and polymerase-associated factor complexes to mediate mono-ubiquitylation of histone H2B on gene bodies of actively transcribed genes[12]. The 1–327 fragment of MED23 also binds to the RunT domain of the transcription factor RUNX2, and this interaction is involved in the regulation of osteoblast differentiation and bone development[9]. Our structure thus rationalizes these previous biochemical results and defines the N-terminal domain of MED23 as an interaction site for partner proteins involved in Mediator functions such as recruitment steps in transcription initiation, splicing and histone post-translational modification on active genes.

**Core MED23 architecture.** Core MED23 contains four different solenoids domains built from HR motifs, that we named 3-HEAT (residues 267–403: 3HR), 5-HEAT (502–751: 5HR), 6-HEAT (752–1034: 6HR) and C-HEAT (1035–1334: 6HR). Their structural organization is illustrated in Fig. 2b−e. Core MED23 also includes a linker region (221–266) which connects 3-HEAT to the N-terminal domain N-HEAT, and a bridge domain which connects 3-HEAT to 5-HEAT (Fig. 1a−d). One face of the 3-HEAT solenoid points towards the center of Core MED23, where it interacts with the bridge domain and 5-HEAT, while the other face of 3-HEAT is solvent exposed (Fig. 2b). 3-HEAT is followed by the bridge domain (residues 404–501), a four helix module (H23-H26) (Fig. 1a) that links 3-HEAT to 5-HEAT and positions these solenoids in a nearly parallel orientation, but with an inverted polarity. The MED23 bridge domain identified here has been shown to be a target for transcriptional regulators. ESX is an Ets transcription factor that is expressed specifically in epithelial cells and that activates the Her2 gene in breast cancers[7]. MED23 binds to the transactivation domain (TAD) of ESX, via a region covering residues 391−487[22], that maps within the bridge domain. Due to its importance in cancerous pathologies, this interaction has been targeted in drug design programs that relied, in absence of structural knowledge about MED23, on the helical

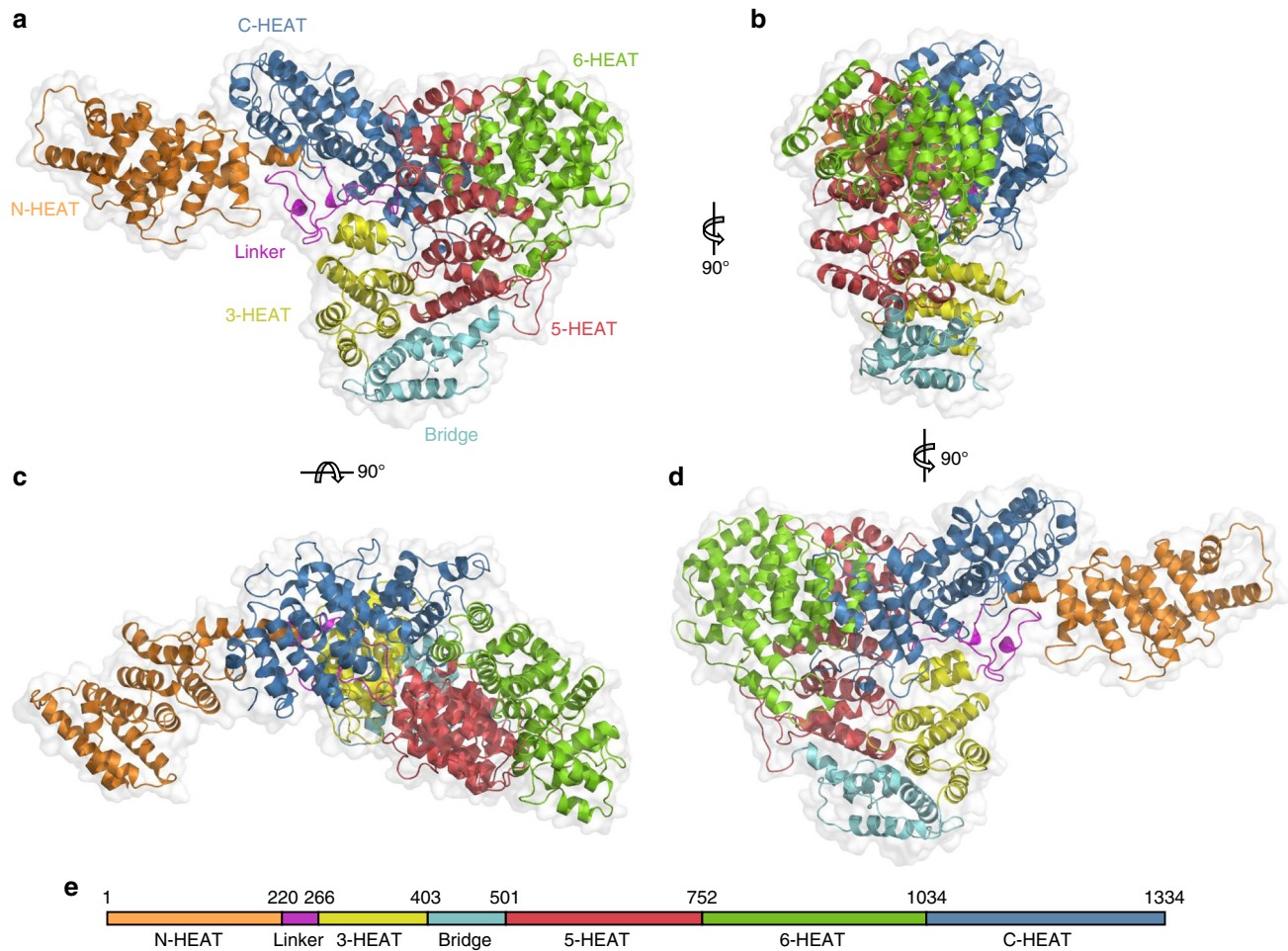

**Fig. 1** Architecture of MED23. **a−d** The overall structure is shown as a translucent surface. The atomic model is shown in ribbon representation, with domains color-coded as follows: MED23 N-HEAT domain, orange; MED23 linker, pink; MED23 3-HEAT domain, yellow; MED23 bridge domain, cyan; MED23 5-HEAT domain, red; MED23 6-HEAT domain, green; MED23 C-HEAT domain, blue. (**a−d**) show different orientations of the complex, as indicated. **e** Schematic of the domain organization of MED23

conformation adopted by the TAD of ESX upon binding to MED23[22–24]. The high resolution structure of MED23 now provides a way to improve drug design programs aiming to control Her2 expression in cancer cells.

Downstream the bridge domain is 5-HEAT that stacks along 3-HEAT but is tilted by ~90° (Fig. 2c). In this orientation, 5-HEAT contacts 3-HEAT mainly via the tips of helices from HR motifs 1 to 4, as well as the loops connecting them. The 5-HEAT solenoid is interrupted between HR motifs 3 and 4 by helix H34 which protrudes towards the center of Core MED23. This face of the 5-HEAT solenoid is solvent exposed on the inner face of the MED23 arch (Fig. 2c), while the other face is buried and packs against 6-HEAT, the next solenoid. 6-HEAT is connected to 5-HEAT by a short linear linker of seven residues. Motifs HR1−HR3 of 6-HEAT do not participate at any interface within Core MED23 (Figs. 2d, 1a, c, d). They curve to guide the three following HR motifs (HR 4–6) of 6-HEAT in front of the HR motifs 2−5 from 5-HEAT. There is an additional HR motif in 6-HEAT, formed by helices H47−H49 and inserted between HR motifs 4 and 5 (Fig. 2d). This motif extends along the 6-HEAT solenoid and contacts helix H27 as well as HR motif 1 and HR motif 2 of 5-HEAT. H48, a short 3₁₀ helix inserted between H47 and H49, is positioned close to the center of Core MED23 (Fig. 2d). Finally, the last solenoid, C-HEAT, completes the roll-up of Core MED23 to contact N-HEAT. C-HEAT connects to and packs against 6-

HEAT via its first HR motif and helix 55, contributing to bend MED23 into its arch-shaped conformation (Fig. 2e). This region also contains H57, a small 3₁₀ helix that is located at the center of Core MED23 in front of helix H48 from 6-HEAT (Fig. 2e). The HR motifs 2−6 in 6-HEAT form a solenoid that structurally resembles N-HEAT (Supplementary Fig. 3). In MED23 only one face of the C-HEAT solenoid is solvent exposed, while the other face stacks onto the linker and N-HEAT (Fig. 2e).

In metazoans, MED23 belongs to the Mediator Tail module, together with MED15, MED16, MED24, and MED25[20]. MED16, MED23, and MED24 are hypothesized to form a submodule[4] which contacts MED14, a subunit that bridges the three Mediator modules[25,26]. How MED23 interacts with these subunits is currently unknown. It has been reported that the N-terminal domain of MED23 binds many partner proteins involved in various Mediator functions, such as hnRNP-L, RNF20/40, or RUNX2[9,10,12], suggesting that N-HEAT is presumably not the primary site of interactions with other subunits of the Mediator complex. We observed that Nb106, the lama nanobody we used to crystallize MED23, can coimmunoprecipitate MED23 incorporated within the Mediator complex (Supplementary Fig. 4a). Nb106 binds via its complementarity determining region (CDR) 3 at an interface within Core MED23, between 5-HEAT, 6-HEAT, and C-HEAT, on one side of the MED23 arch (Supplementary Fig. 4b). In addition, a commercial antibody

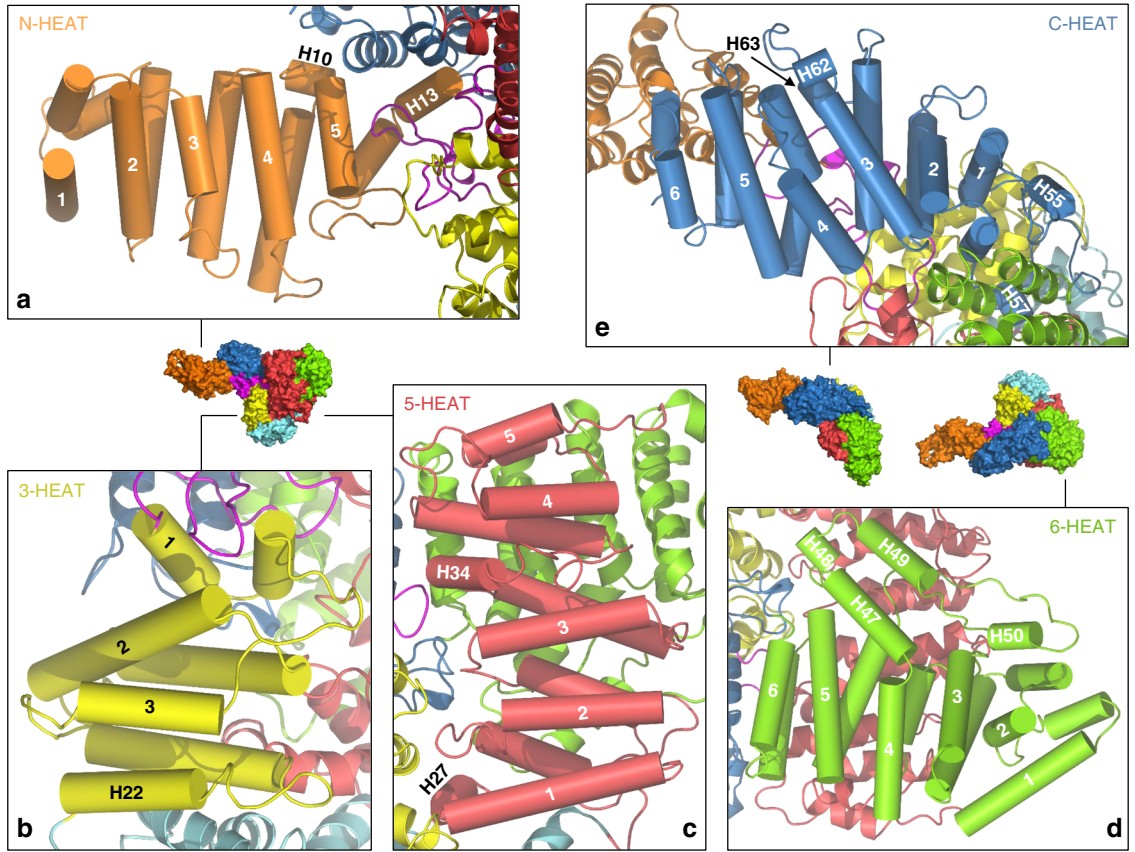

**Fig. 2** Structure of MED23 solenoid domains. **a** N-HEAT domain, with the five HR motifs labeled. H10 and H13, which pack against C-HEAT and, for the later, connects N-HEAT to the linker, are indicated. **b** 3-HEAT domain, with three HR motifs. H22, which connects 3-HEAT to the bridge domain, is indicated. **c** 5-HEAT domain, with five HR motifs. H34 disrupts the solenoid between HR motifs 3 and 4 and is indicated. HR motifs 2−5 are positioned in front of HR motifs 4−6 from the 6-HEAT domain, in green in the background. **d** 6-HEAT domain, in green, with six HR motifs. The insertion between HR motifs 4 and 5 also contains an HR motif (helices 47 and 49) encompassing helix H48, positioned at the center of Core MED23. These three helices cover HR motifs 1 and 2 as well as helix H27 from 5-HEAT. **e** C-HEAT, with six HR motifs. HR motif 1 stacks against the last HR motif from the 6-HEAT domain. HR motifs 2−6 form a solenoid closely resembling that of N-HEAT. Color code is as in Fig. 1. The orientation of MED23 with respect to Fig. 1 is also represented

targeting a region covering amino acids 400 to 450 has been reported to coimmunoprecipate Mediator[12] (Supplementary Fig. 4b). It thus appears that the regions or interfaces targeted by these antibodies are not involved in interactions with other Mediator subunits. Collectively, these data suggest that, within Core MED23, the concave (Fig.1a) and convex (Fig. 1d) faces of the MED23 arch are available to interact with other Mediator subunits, in addition to transcriptional regulators.

On the concave face of the MED23 arch, the linker, 3-HEAT, 5-HEAT, and C-HEAT interact and form a complex interface (Fig. 3a). In this interface 5-HEAT is packed along 3-HEAT and the linker, while C-HEAT contributes helices H64, H65, and H67 which interact with the linker and the fourth HR motif of 5-HEAT (Fig. 3a). On the convex face of the MED23 arch, 6-HEAT and C-HEAT interact respectively via their C and N terminal regions (Fig. 3b). These two solenoid domains stack against 5-HEAT and 3-HEAT, creating a funnel-shaped surface bordered by helices from the four domains (Fig. 3b). The bridge domain contributes to both faces of the arch with helices H23, 25, and 26 contacting 3-HEAT and 5-HEAT (Fig. 3a, b). The MED23 arch displays specific structural features. On the concave face, an extended groove runs along the surface between the linker, C-HEAT, 3-HEAT, and 5-HEAT (Fig. 3a). This groove is well exposed, and its accessibility is only slightly limited by the V230-S240 segment of the linker, which is variable among metazoans

(Supplementary Fig. 5). The groove defines a positively charged surface propagating from N-HEAT to the center of the concave face. It communicates with the convex face of core MED23 via a buried negatively charged cavity (Supplementary Fig. 6) that joins the funnel-shaped surface, close to helices H48 and H57 (6-HEAT and C-HEAT, respectively). Finally, on the convex face, we observed a partly exposed negatively charged groove near helices H16 and 19 (3-HEAT), H23 (Bridge), and H21 and H27 (5-HEAT) (Fig. 3b and Supplementary Fig. 6). These sequence motifs are highly conserved across metazoans, demonstrating their important structural and/or functional roles (Supplementary Fig. 5). Altogether, they provide very specific interfaces which might be involved in the recruitment of other Mediator subunits and/or the binding of specific transcription factors. They also reveal signaling transmission paths that link all domains within MED23 and connect both faces of the MED23 arch. The presence of buried charges also suggest that MED23 may adopt a dynamical behavior upon effector binding, which could result in important structural changes. Although these molecular properties have still to be deciphered, data agree with the implication of these interfaces in important MED23 functions. It has been shown that the R611Q mutation in MED23 is linked to neurological disorders, highlighting its role in brain development and intellectual diseases[27]. This mutation specifically impairs the response of JUN and FOS immediate early genes to serum

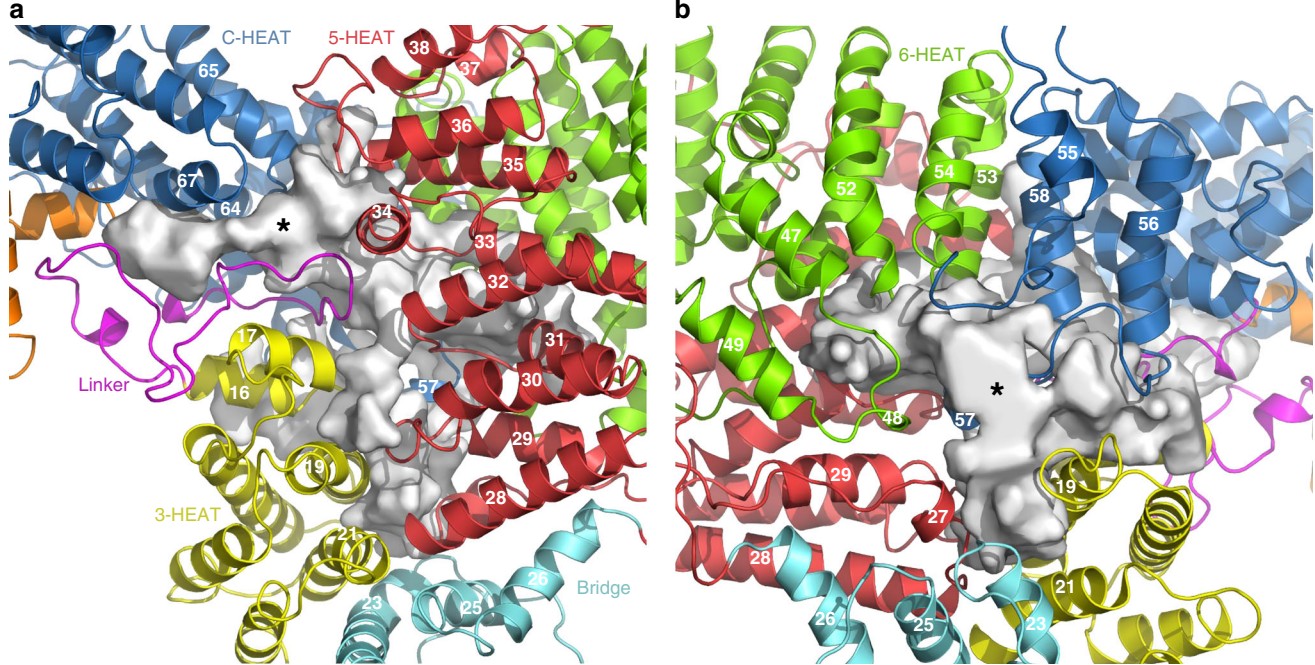

**Fig. 3** Illustration of extended grooves in Core MED23. View of the interface on the concave (**a**) and convex (**b**) faces of Core MED23. The linker, 3-HEAT, Bridge, 5-HEAT, 6-HEAT, and C-HEAT are represented in ribbons, following the color code as in Fig. 1. **a** On the concave face of the MED23 arch, the extended groove running along the surface is illustrated, with the * pointing to the opening of the cavity. **b** On the convex face of the arch, 6-HEAT and C-HEAT stack against 3-HEAT and 5-HEAT, defining a funnel-shaped groove. Here also the * represents the opening of the cavity. Helices contributing to the interfaces are labeled

mitogens. In our structure, R611 rims the funnel on the convex face of MED23 and engages in a buried salt bridge with E565 (Fig. 4), an interaction which is conserved in metazoans. R611 and E565 are stacked against an aromatic cluster at the center of Core MED23 involving H613 and 614 (5-HEAT), H911 and W912 (H48, 6-HEAT), F1094 (H57, C-HEAT) and Y278 (H16, 3-HEAT) (Fig. 4). This aromatic cluster surrounds a signature motif composed of residues D1091, W1092, R1093, E1096 (helix H57), and D529 (Fig. 4), that is conserved in metazoans. The aromatic cluster also contacts the concave face of MED23 via H614 and Y278 at the depth of the groove on this face (Fig. 4). The R611 mutation presumably impacts these structural features, thus affecting both interfaces of the MED23 arch. Although these residues are not fully exposed in the structure to engage in direct interactions with partner proteins, they might play a crucial role in contributing to the functionality of the core MED23 interfaces. The MED23 structure thus provides a molecular rationale to further explore the precise downstream factors in the serum response pathway leading to intellectual disability.

The structural characterization of MED23 reported here represents an essential step to understand at the molecular level how this subunit regulates transcription via many factors including ELK1, ESX, RUNX2, IRF7, and how it participates to postrecruitment events in elongation, splicing, and post-translational modification processes on active genes. It also paves the way to the challenging structural characterization of the metazoan Mediator Tail module, which remains mostly unresolved.

## Methods

**Recombinant MED23 protein production**. DNA sequence encoding human MED23 isoform 1 (NM_004830.3) was amplified by PCR and cloned into a pFastBac that contains a rhinovirus 3C protease site followed by a 6xHis tag to generate a C-terminal 6xHis-tagged MED23 protein. Recombinant baculovirus were obtained using the Bac-to-Bac expression system (Invitrogen) as specified by the manufacturer. Seventy-two hours after baculovirus infection, the SF21 cells

were resuspended in MEDlysis buffer containing 50 mM Tris/HCl pH 7.5, 300 mM NaCl, 1 mM TCEP, 10% (w/v) glycerol, 0.1% Igepal CA-630 and 5 mM Imidazole. After centrifugation, the purification of MED23 was performed on a Talon Resin column (Clontech). The beads were washed with MEDlysis buffer and MED23-6His was eluted using the MEDlysis buffer supplemented with 250 mM Imidazole. The pooled fractions were dialyzed overnight at 4 °C against 20 mM Tris/HCl pH 7.5, 60 mM NaCl, 1 mM TCEP and 10% (w/v) glycerol and further purified by size-exclusion chromatography on a Superdex 200 column (GE Healthcare) pre-equilibrated with 20 mM Tris/HCl pH 7.5, 100 mM NaCl, 1 mM TCEP and 10% (w/v) glycerol. The MED23-6His protein was then concentrated to 4 mg ml$^{-1}$ and aliquots were flash-frozen and stored at −80 °C.

**Generation, expression and purification of nanobody Nb106**. Nanobodies targeting MED23 were generated following established protocols[28]. The plasmids containing C-terminal His6-tagged Nb were transformed into *E. coli* strain WK6, grown in TB medium containing 0.1% glucose, 2 mM MgCl$_2$ and 100 mg ml$^{-1}$ ampicillin at 37 °C until the A600 nm of the sample reached 1.2, and then induced with 1 mM IPTG for 4 h. Cells were collected and the periplasmic fraction was extracted using the modified osmotic shock protocol. The periplasmic extract containing Nb was supplemented with 10 mM imidazole and incubated with Ni-NTA agarose (Qiagen) for 1 h at 4 °C. The beads were washed with 50 column volumes of PBS containing 10 mM imidazole, and Nb were eluted from the resin with a PBS buffer containing 300 mM imidazole. The eluted fractions containing Nb were concentrated and injected into a Superdex 75 (GE Healthcare) size-exclusion chromatography column pre-equilibrated with 20 mM Tris/HCl pH 7, 150 mM NaCl, 1 mM TCEP. Nb were concentrated to 4 mg ml$^{-1}$, flash-frozen and stored at −80 °C. The nanobody Nb106 successfully crystallized MED23.

**Crystallization, data collection, and structure determination**. Purified MED23 was mixed with Nb106 in a 1:1.2 ratio in order to obtain final protein concentrations of respectively 4 and 0.4 mg ml$^{-1}$. The protein mixture was then incubated for 1 h at 4 °C before crystallization trials. Native MED23 crystals were obtained at 20 °C using the hanging drop vapor diffusion method. Crystals were grown in 10% PEG6000 and 100 mM HEPES buffer (pH 7.0). Native crystals were collected and transferred directly into the cryo-solution corresponding to the mother liquor supplemented with 20% glycerol, and subsequently flash-frozen in liquid nitrogen. Orthorhombic crystals were obtained in all crystallization drops, with the exception of one drop where two crystal forms were obtained simultaneously. In addition to the orthorhombic crystals, a few monoclinic crystals appeared in this drop, one of which diffracted to ~2.65 Å resolution (Supplementary Table 1). Both crystal forms contain one complex of MED23 and Nb106 in the asymmetric unit.

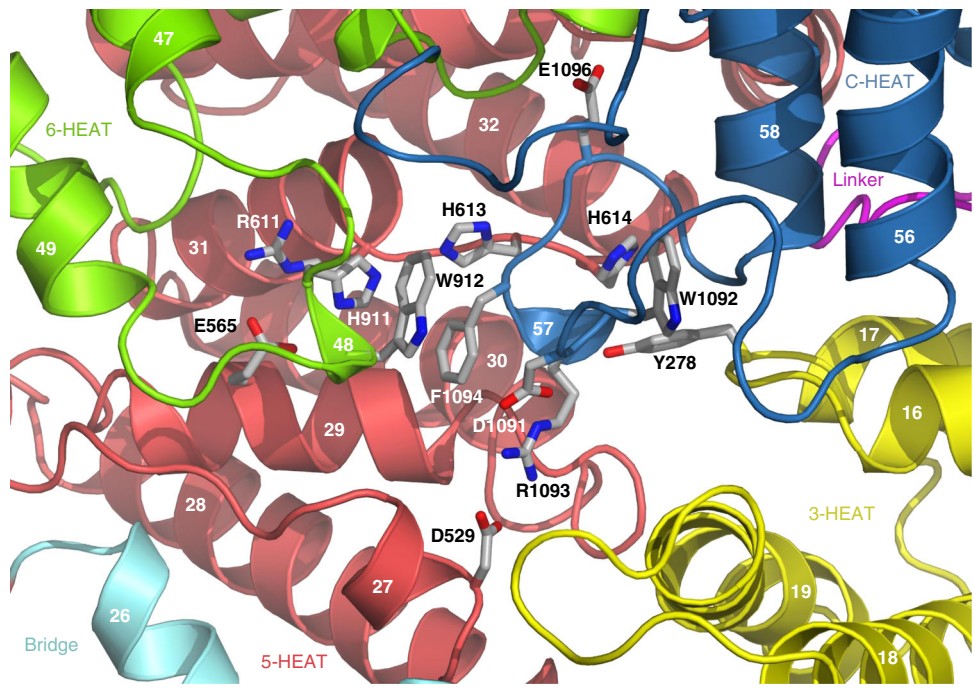

**Fig. 4** Interface between 3-HEAT, 5-HEAT, 6-HEAT, and C-HEAT domains. Structural illustration showing the molecular environment at the interface between 3-HEAT, 5-HEAT, 6-HEAT, and C-HEAT on the convex face of core MED23. R611 (corresponding to R617 in isoform 2, NM_015979.2) forms a buried salt bridge with E565 and is stacked against an aromatic cluster surrounding structural elements strictly conserved in all metazoans, at the center of the funnel

For structure determination, native orthorhombic crystals were derivatized. Pt and Au derivatives were obtained by soaking native crystals for 2 min in a solution containing 20% glycerol, 10% PEG6000, 100 mM HEPES buffer (pH 7.0) and 45 mM $K_2PtCl_4$ or 45 mM KAu(CN)$_2$, respectively.

Data were collected at 100K at ESRF and SOLEIL synchrotrons in France (Supplementary Table 1). All data were processed with XDS and XSCALE[29]. Crystals used for phasing and building the initial models were from the orthorhombic forms, which diffracted to resolutions between 3.0 and 3.85 Å. The structure was solved with the SHARP/autoSHARP distribution[30] by the multiple isomorphous replacement method with anomalous scattering using the platinum and gold derivatives (Supplementary Table 1). A poly-ala model was manually built with Coot[31], corresponding to 80% of the 71 helices, although with errors in polarity. Rounds of refinement at 3.0 Å resolution combining the use of Refmac[31] and Buster[32] allowed to obtain improved maps, to identify and fit a nanobody (PDB code 5IMK) related to Nb106, which progressively helped to correct and complete the poly-ala MED23 model. We then placed the complex formed by MED23 and Nb106 in the monoclinic crystal form using MOLREP[31], which allowed to complete the poly-ala MED23 model and assign the sequence, from residues 1 to 1334. The last steps of refinement were performed at 2.8 Å resolution using PHENIX[33], which allowed us to better control the geometry of the model by optimizing the X-ray/stereochemistry weight. The high resolution limit was set up to 2.8 Å resolution to prevent overfitting by using diffraction data with I/σI greater than 2. No water molecule was added to the model, as we did not observe any obvious electron density corresponding to water molecules and their forced addition by PHENIX[33] resulted in a worsening $R_{free}$. The final model is complete, with the exception of the last 30 residues, which have no visible electron density. The structure is well resolved, with only a few sequence segments from loops which are poorly ordered (N233-I237; N290-R296; E316-S327; G364-L367; Y403-Y408, D439-Q444; N502-R506; Q940-D942; N1054-E1058; E1182-G1185). These segments were kept in the model to ensure continuity of the chain. The final model has an R and $R_{free}$ of 19.9 and 24.0%, respectively. The geometry of the final model agrees well with structures refined to 2.8 Å resolution, with 94.56 and 4.89% of residues respectively in the favored and allowed regions of the Ramachandran plot, and 0.55% of outliers. The cavities in MED23 which are described and illustrated in Fig. 3 have been analyzed using POCASA[34], and rendered using PyMol[35].

**Coimmunoprecipitation experiments.** HEK-293 human cells were lysed in a buffer containing 50 mM Tris/HCl pH 8, 250 mM NaCl, 1 mM EDTA, 1 mM TCEP, 10% glycerol, 3 mM $MgCl_2$ and 0.1% Igepal CA-630. After centrifugation, extracted proteins were incubated with 2 µg of Nb106 overnight at 4 °C. The resulting Nb106 immuno-complexes were then fished with 50 µl of CaptureSelect™ C-tag Affinity Matrix (Thermo). After extensive washing, the precipitated proteins were identified by western blot. The antibodies used to detect Mediator subunits were from Santa Cruz (anti-Med7 sc-12457 and anti-Cdk8 sc-1521), BD

Pharmigen™ (anti-Med23 550429) or from Bethyl Laboratories (anti-Med1 A300–793A, anti-Med14 A301–044A, anti-Med16 A303–668A, anti-Med18 A300–777A, and anti-Med24 A301–472A). The same blot was successively hybridized with different primary antibodies combinations at 1:1000, Med1-Med23-Med16-Med24, Med14, Cdk8-Med7 and Med18, respectively. Western blot was performed using an ECL kit (Amersham GE Healthcare Life Sciences) based on the manufacturer's recommendations. HRP-conjugated secondary antibodies were used at 1:10,000 and were purchased from Amersham GE Healthcare Life Sciences. All the membranes were scanned on an ImageQuant LAS 500 (GE Healthcare Life Sciences).

**Data availability**. Coordinates and structure factors for MED23 have been deposited in the protein data bank under accession 6H02. Other data are available from the corresponding authors upon request.

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

## Acknowledgements

D.M., B.C., F.D., Z.L., A.V. and V.V. are researchers from the Centre National de la Recherche Scientifique in France (CNRS). This work was supported by the French National Research Agency (Grant ANR-DS0401-2016 MEDNET). We thank INSTRUCT, part of the European Strategy Forum on Research Infrastructures (ESFRI) and the Hercules Foundation Flanders for their support to the Nanobody discovery. We thank Nele Buys for the technical assistance during Nanobody discovery. We thank Ahmed Haouz from the Pasteur Institute in Paris for his help during crystallization trials. Access to synchrotrons has been achieved through the Beam Allocation Group (BAG) LOR. We thank Leonard Chavas, Serena Sirigu, William Shepard, and Martin Savko at beamlines Proxima-1 and Proxima-2 at the synchrotron SOLEIL and Carlo Petosa and Daniele de Sanctis at beamlines ID23 and ID29 at the European Synchrotron Radiation Facility, Grenoble for their assistance during data collection. B.C. and V.V. are indebted to the Research Federation FRABio (Univ. Lille, CNRS, FR 3688, FRABio, Biochimie Structurale et Fonctionnelle des Assemblages Biomoléculaires) for providing the scientific and technical environment conducive to achieving their contribution.

## Author contributions

D.M. and B.C. designed, performed experiments and determined the structure. P.R., F.D., Z.L. and A.V. provided technical support. E.P. and J.S. generated nanobodies. D.M., B.C., P.R., A.V. and V.V. designed experiments, analyzed data and wrote the manuscript.

## Additional information

**Competing interests:** The authors declare no competing interests.

