## [Peer Review File · Nature Communications]

Reviewers' comments:

Reviewer #1 (Remarks to the Author):

In the manuscript titled "Architecture of the Human Mediator subunit MED23", Monte et al., have determined the structure of MED23 in complex with a stabilising lama nanobody. As explained by the authors, this is the first report of structural information on the tail module of the Mediator complex, a central coactivator of transcription in eukaryotic cells. The structure is important as it is a starting point to provide insights into the cross-talk between Mediator and transcriptional activators reported previously. The authors have used the nanobody to delineate the architecture and the interaction surfaces of MED23 within the Mediator complex through co-immunoprecipitation. The structural details of MED23 are well explained. The manuscript makes a valuable contribution to our understanding of the structure of the Mediator tail module. I highly recommend publication of this fine work after the following concerns have been addressed.

1. The difference between R_{work} and R_{free} is relatively high. The authors should see whether they overfitted the data. They may consider truncating the data to a slightly lower resolution limit or could carefully check the solvent.
2. In Extended data Table 1, details of the values from the Ramachandran plot should be included. The legend should explain what the values in parantheses represent as well as mention the formula used to calculate R_{meas} .
3. The authors may consider labelling the domains in all the cartoons representing the structure.
4. Line 531. Table 1 should include the number of molecules/ASU.
5. Line 53. The author should comment more on the crystal packing and how the nanobody helped in crystallization.
6. The authors should reconsider why figure 3 is required and better explain this to the reader.
7. Extended data Fig 1. Why is there no water molecules in the 2Fo-Fc map? Maybe display a different part of the molecule.

8. Extended data Fig 4. Please label the protein marker.

9. Extended data Fig 6. Please include the unit for the electrostatic potential map.

10. In Extended data Table 1 the authors have to specify whether these are unique or total reflections and how this relates to the redundancy provided.

11. The authors generally did a good job in referring to the relevant literature but because there are many more papers on Mediator structure that do not have to be cited here but are of interest to reader who enter the field, I recommend to include the recent review on Mediator structure by Plaschka et al., JMB 2016, which contains also a table of yeast and human Mediator subunits and points to all Mediator structural literature before the most recent papers that the authors have cited. This is optional of course.

Reviewer #2 (Remarks to the Author):

Major remarks:

Mediator complex is key transcriptional regulator. Authors present crystal structure of human Mediator subunit Med23 at 2.65 Å. hMed23 was co-crystallized with nanobody against hMed23.

Even though the authors go to great lengths to describe their solved structure and this structure does add to the understanding of the Mediator-tail to some extent, functional data to support the solved structure as well as to enable a better understanding of the tail-module architecture or interaction with other factors is, nevertheless, lacking. For example, structure based-analysis of conserved residues, as has been performed (see extended data figure 5), could be used to identify binding sites of other tail subunits via computational, biochemical, in-vivo or other methods. Similarly, factors like E1A, RNF20/40, RUNX2, hnRNP-L have already been identified as binding partners of the Med23 N-terminus. Some of those, if the interaction is well- or re-validated, could be further studied in a structure-based mutational analysis to gain insight into the mode of binding and perhaps regulation.

Minor remarks:

Authors failed to cite the paper: Imasaki et al (2011), which was indeed the first high-resolution structure of the sub-complex of mediator from yeast. This is a gross oversight. In addition, even though we all assume evolutionary conservation of the mediator subunits across the species,

authors must better articulate the origins of species from which mediator or its sub-complexes came. For example, X-ray structures of the head were from *S. cerevisia* and those of the middle were from *S. pombe*.

Given the resolution (2.65 Å) of the structure, R-values (20.6/26.4) are relatively high, suggesting the possibility that refinement did not go well despite of 2.65 Å datasets. It is possible that the MED23 crystals may be twinned? It is not clear why authors described it “architecture” if the resolution is 2.65 Å?

The data authors show in the extended data figure 4a, that binding of the nanobody does not interfere with the integrity of the Mediator complete is inadequate for the following reasons: 1) the show data does not show that Nb106 does immunoprecipitated the primary target Med23; 2) Input-signal does not align with co-immunoprecipitated band corresponding to Cdk8, Med14 and Med16; 3) sizemarker is not labeled; 4) overall labeling of image a is insufficient in figure legends. (what does (i); (+) and (-) refer to?)

Reviewer #1 (Remarks to the Author)

In the manuscript titled "Architecture of the Human Mediator subunit MED23", Monte et al., have determined the structure of MED23 in complex with a stabilising llama nanobody. As explained by the authors, this is the first report of structural information on the tail module of the Mediator complex, a central coactivator of transcription in eukaryotic cells. The structure is important as it is a starting point to provide insights into the cross-talk between Mediator and transcriptional activators reported previously. The authors have used the nanobody to delineate the architecture and the interaction surfaces of MED23 within the Mediator complex through co-immunoprecipitation. The structural details of MED23 are well explained. The manuscript makes a valuable contribution to our understanding of the structure of the Mediator tail module. I highly recommend publication of this fine work after the following concerns have been addressed.

1. The difference between R_{work} and R_{free} is relatively high. The authors should see whether they overfitted the data. They may consider truncating the data to a slightly lower resolution limit or could carefully check the solvent.

Answer. We initially set up the high resolution cutoff to diffraction data with $I/\sigma I \sim 1.2$, which is a standard adopted nowadays by many in the community. Although we believe there is nothing wrong with this approach, we agree that it can be assumed that the true resolution is not as good. We modified the cutoff in such a way to include data with $I/\sigma I > 2$ (a much safer cutoff), which resulted in a 2.8 Å cutoff.

Due to concerns of the reviewers about the refinement, we also carefully checked the geometry of the model. We turned to PHENIX to finalize the refinement, as PHENIX helped us to better control the refinement procedure regarding the geometry of the model. Cycles of refinement and close inspection of the electron density map allowed to obtain a model with a good geometry at 2.8Å resolution (see pdb report), as well as to lower the R and R_{free} to values of 19.9% and 24.0%, respectively, without water molecules (previously, we had 20.6% and 26.4%, with 172 water molecules). Thus the difference between R and R_{free} is now only 4%, and with 24% the R_{free} is in the best ones observed for a 2.8 Å resolution structure, with a very good model geometry. No water molecule was added to the model, as we did not observe any obvious electron density corresponding to waters (round electron density within hydrogen bond distance of appropriate chemical groups of residues) and forced addition of waters by PHENIX resulted in a worsening R_{free} . Of note, using standard parameters, PHENIX does not search for water molecules at the 2.8Å resolution limit. We want to thank the reviewer for catching our attention on this parameter, because we had probably overinterpreted the solvent contribution at this resolution. The geometry of the final model agrees well with structures refined to 2.8Å resolution, with 94.56 % and 4.89 % of residues respectively in the favored and allowed regions of the Ramachandran plot, and 0.55% of outliers which mostly belong to disordered loops. We choose to keep the few disordered loops within the model, to ensure continuity of the chain, which is much more convenient for the community to manipulate the structure.

The methods section has been modified and completed to describe how the refinement was finalized.

2. In Extended data Table 1, details of the values from the Ramachandran plot should be included. The legend should explain what the values in parantheses represent as well as mention the formula used to calculate R_{meas} .

Answer. Values from the Ramachandran plot have been included in the Methods section (we choose to indicate them in the text rather than in Table 1, to follow Nature Comm guidelines). The legend now explains what the values in parentheses represent, and the formula used to calculate R_{meas} has been added in footnote.

3. The authors may consider labelling the domains in all the cartoons representing the structure.

Answer. The labelling of the domains has been included.

4. Line 531. Table 1 should include the number of molecules/ASU.

Answer. The number of molecules/ASU has been added in the Methods section.

5. Line 53. The author should comment more on the crystal packing and how the nanobody helped in crystallization.

Answer. We now describe the interface between the nanobody and MED23 and the contribution of the nanobody to crystal packing.

6. The authors should reconsider why figure 3 is required and better explain this to the reader.

Answer. Figure 3 illustrates very important features of the MED23 structure that link all the HEAT solenoids together and may reflect putative binding sites for other Mediator subunits and transcription factors. The text has been modified to more clearly specify these points.

7. Extended data Fig 1. Why is there no water molecules in the 2Fo-Fc map? Maybe display a different part of the molecule.

Answer. We choose to illustrate the electron density of this part of the asymmetric unit because it shows the interface between MED23 and the nanobody Nb106. There was no water molecule in this interface. The question of waters in the structure was a very pertinent one and has been addressed above.

8. Extended data Fig 4. Please label the protein marker.

Answer. The size of the protein marker has been included.

9. Extended data Fig 6. Please include the unit for the electrostatic potential map.

Answer. The Unit ($kT e^{-1}$) is now indicated in the figure.

10. In Extended data Table 1 the authors have to specify whether these are unique or total reflections and how this relates to the redundancy provided.

Answer. Unique reflections is now specified in Table 1.

11. The authors generally did a good job in referring to the relevant literature but because there are many more papers on Mediator structure that do not have to be cited here but are of interest to reader who enter the field, I recommend to include the recent review on Mediator structure by Plaschka et al., JMB 2016, which contains also a table of yeast and human Mediator subunits and points to all Mediator structural literature before the most recent papers that the authors have cited. This is optional of course.

Answer. References have been added to cite most (if not all) the structural breakthroughs regarding the structure of Mediator.

Reviewer #2 (Remarks to the Author):

Major remarks:

Mediator complex is key transcriptional regulator. Authors present crystal structure of human Mediator subunit Med23 at 2.65 Å. hMed23 was co-crystallized with nanobody against hMed23.

Even though the authors go to great lengths to describe their solved structure and this structure does add to the understanding of the Mediator-tail to some extent, functional data to support the solved structure as well as to enable a better understanding of the tail-module architecture or interaction with other factors is, nevertheless, lacking. For example, structure based-analysis of conserved residues, as has been performed (see extended data figure 5), could be used to identify binding sites of other tail subunits via computational,

biochemical, in-vivo or other methods. Similarly, factors like E1A, RNF20/40, RUNX2, hnRNP-L have already been identified as binding partners of the Med23 N-terminus. Some of those, if the interaction is well- or re-validated, could be further studied in a structure-based mutational analysis to gain insight into the mode of binding and perhaps regulation.

Answer. Prior to our study nothing was known about the MED23 structure, not even the fact that it comprises HEAT repeat-like motifs. Now with the structure in hands, we agree that the next move will be to characterize the interaction of MED23 with other Mediator subunits and with transcription factors. Unfortunately, this is not a straightforward task. Regarding other Mediator subunits, MED23 has been reported to belong to the Tail module. However, so far, it has not been possible to reconstitute in vitro this submodule and how its subunits interact is yet currently unknown, despite ~15 years of research. So the next step to better understand the Tail module architecture is to try to reconstitute it in vitro and then probe it functionally and/or structurally. The structures of other subunits of the Tail (MED24, MED16, MED15) are currently unknown and, in this regard, our study paves the way to such studies, as pinpointed by reviewer 1. We searched for putative interactions sites in MED23 by *in silico* methods that resulted in a huge number of predicted sites (~30) that mapped all along the surface of MED23. The larger putative interaction sites were predicted in the grooves running between the core MED23 domains on both the concave and convex faces of MED23 (which are illustrated in Figure 3). Probing these grooves by site-directed mutagenesis is not currently an attainable task for us, because these grooves involve more than two hundred residues that are also often conserved for structural reasons (interfaces between HEAT solenoid domains). We have modified the manuscript to pinpoint these facts.

For other factors like E1A, RNF20/40, RUNX2, or hnRNP-L, these interactions must be validated before they can be probed in vitro. Regarding RNF20/40, RUNX2, or hnRNP-L, these interactions have been proposed by the Gang Wang's lab by co-ip experiments using truncated MED23 proteins. In absence of any structural information about MED23, these mappings have been performed using randomly truncated MED23 proteins, and must be revalidated. Supplementary Figure 5 shows that the Nter is poorly conserved within metazoans, excepting one cluster of residues which is at the junction of the Nter with core MED23 (charged cluster between N-HEAT and C-HEAT domains) and which are conserved (also) for structural reasons. To our knowledge, the regions of MED23 that interact with E1A have not been mapped.

Minor remarks:

Authors failed to cite the paper: Imasaki et al (2011), which was indeed the first high-resolution structure of the sub-complex of mediator from yeast. This is a gross oversight. In addition, even though we all assume evolutionary conservation of the mediator subunits across the species, authors must better articulate the origins of species from which mediator or its sub-complexes came. For example, X-ray structures of the head were from *S. cerevisia* and those of the middle were from *S. pombe*.

Answer. We apologize for this omission due to the reference number limit in the first submitted version of the manuscript. References have been added to cite most (if not all) the structural breakthroughs regarding the structure of Mediator.

Given the resolution (2.65 Å) of the structure, R-values (20.6/26.4) are relatively high, suggesting the possibility that refinement did not go well despite of 2.65 Å datasets. It is possible that the MED23 crystals may be twinned? It is not clear why authors described it "architecture" if the resolution is 2.65 Å?

Answer. Thank you for this concern, also raised by reviewer #1. Answer regarding the Rfree is provided above. MED23 crystals are not twinned, this is shown in the pdb report with the analysis by Xtriage (4 Data and refinement statistics). Finally, we changed the title from "Architecture..." to "Crystal structure..." to better reflects the resolution achieved, 2.8 Å now.

The data authors show in the extended data figure 4a, that binding of the nanobody does not interfere with the integrity of the Mediator complete is inadequate for the following reasons: 1) the show data does not show that Nb106 does immunoprecipitated the primary target Med23; 2) Input-signal does not align with co-immunoprecipitated band corresponding to Cdk8, Med14 and Med16; 3) sizemarker is not labeled; 4) overall

labeling of image a is insufficient in figure legends. (what does (i); (+) and (-) refer to?)

Answer. A new Supplementary Figure 4a is now presented. The new data now clearly show that Nb106 immunoprecipitates MED23. Anti-Med1 has also been added. The molecular weight marker is now indicated and the figure legends have been updated to indicate what does I, + and – refer to. In our previous figure 4a, input sample was highly concentrated, which may explain why it did not align well with Cdk8, Med14 and Med16. In the current figure, input lane now corresponds to 1/100 of the lysate used for Co-IP.